# Sub- and Near-Critical Hydrothermal Carbonization of Animal Manures

Kyoung S. Ro [1,*], Michael A. Jackson [2], Ariel A. Szogi [1], David L. Compton [2], Bryan R. Moser [2] and Nicole D. Berge [3]

1   USDA Agricultural Research Service, Coastal Plains Soil, Water & Plant Research Center, 2611 W. Lucas St., Florence, SC 29501, USA; ariel.szogi@usda.gov
2   USDA Agricultural Research Service, National Center for Agricultural Utilization Research, 1815 N. University St., Peoria, IL 61604, USA; michael.jackson@usda.gov (M.A.J.); david.compton@usda.gov (D.L.C.); bryan.moser@usda.gov (B.R.M.)
3   Department of Civil and Environment Engineering, University of South Carolina, Columbia, SC 29208, USA; berge@engr.sc.edu
*   Correspondence: kyoung.ro@usda.gov; Tel.: +1-843-669-5203

**Abstract:** To produce hydrochar with less volatile matter (VM) and more fixed carbon (FC) to increase its stability, this study compared the hydrothermal carbonization (HTC) of hen (HM) and swine (SM) manures at typical HTC sub-critical temperature of 210 °C and slightly super-critical temperature of 400 °C. Physico-chemical properties such as proximate analysis; ultimate analysis; Brunauer–Emmett–Teller (BET) surface area; higher heating value (*HHV*); chemical oxygen demand (*COD*); and inorganic nutrients of hydrochar, gaseous, and liquid products were determined. As expected, both VM and yield decreased with temperature. The heats of HTC reactions were estimated to be exothermic, ranging from −5.7 to −8.6 MJ/kg. The FC approximately doubled, while VM significantly decreased with a yield of 42.7%, suggesting the high potential of producing more stable hydrochar via near-critical HTC (NCHTC) treatment of SM. Additional work is needed before recommendations on carbonization temperatures can be made. Specifically, there is a need to experimentally investigate how the chars produced from each carbonization condition influence plant growth and soil emissions.

**Keywords:** hydrothermal carbonization (HTC); near-critical HTC; animal manure

## 1. Introduction

Hydrothermal carbonization (HTC) typically uses subcritical water with hydrothermal reaction temperatures ranging from 150 to 270 °C to convert biomass feedstock mainly into a solid carbonaceous product called hydrochar [1]. The hydrochars made from HTC treatment of animal manures have been evaluated for agricultural, energy, and environmental applications [2–5]. When swine-manure-based hydrochar was applied to soil as an amendment, the hydrochars improved soil fertility by providing abundant nutrients to plants yet reducing water-polluting potential by not releasing these nutrients in leachate [2]. Manure-based hydrochars also showed remarkable ability to remove both polar and non-polar organic pollutants in water via adsorption. In contrast, biochars made of the same manure feedstock were adequate only for non-polar compounds [4]. In addition, carbonization at temperatures as low as 150 °C was sufficient to result in both total pathogen kill and complete removal of microbially derived DNA from animal carcasses [6].

One of the shortfalls of hydrochar, when applied to soil to sequester soil carbon, is its relatively lower stability than biochar (hereafter referred as pyrochar). This lower stability is due to their lower degree of carbonization and aromatization with more alkyl and carboxylate groups than pyrochar [7]. Marco et al. (2016) reported that 23–30% of hydrochar-C was mineralized after 19 months of field incubation compared to non-significant mineralization of pyrochar-C. As a result, the mean residence times for the hydrochar ranged from

3 to 14 years compared to 16 to 224 years for the pyrochar. Consequently, the hydrochar-amended soil emitted significantly more $CO_2$, with much of the C from hydrochar, while pyrochar-amended soil decreased $CO_2$ emissions [8].

The stability of hydrochar in soil may be related to its chemical oxidation resistance potential. The chemical oxidation resistance of hydrochar carbonized at 250 °C was significantly lower than that of pyrochar pyrolyzed at 600 °C. Still, it was comparable to that of pyrochar pyrolyzed at 450 °C or higher than the pyrochar made at 250 °C [9]. The stability of pyrochar increased with pyrolysis temperature. Similarly, the H/C ratio of hydrochar decreased with increasing HTC temperature, indicating more aromatization of hydrochar [10]. Therefore, the stability of hydrochar may increase with HTC temperature. However, as the HTC temperature increased, hydrochar yield decreased, significantly at temperatures much higher than the critical point of water (i.e., 374 °C and 22 MPa).

When biomass feedstock is subjected to temperatures higher than 600 °C, most the feedstock C is converted to energy gases with meager yields of hydrochar. Therefore, it is referred to as supercritical water hydrothermal gasification (HTG) [11]. Cao et al. reported that chicken manure was almost entirely gasified at 620 °C without catalysts [12]. In contrast, Madenoglu et al. found that the C yield in the solid residue ranged from 43.7 to 56.7% for hydrothermal gasification of model compounds cellulose and lignin alkali conducted at 400 °C without using catalysts [13]. To promote hydrothermal gasification below 600 °C, but near the critical point, catalysts are usually used to produce energy gas similar to non-catalytic HTG at higher temperatures. Youssef et al. evaluated different catalysts for their effectiveness in $H_2$ production and chemical oxygen demand (*COD*) reduction from HTG of hog manure at 500 °C [14]. Palladium catalysts supported by activated carbon (Pd/AC) were the most effective at producing $H_2$, while catalytic NaOH resulted in the largest reduction of *COD*. Methane and $CO_2$ were the main gas products when gasified near critical point with a ruthenium catalyst [15]. Nanda et al. reported that HTG of horse manure produced gas with higher energy value when gasified at temperatures (400 to 600 °C) or with the use of catalysts [16]. The decrease in H/C and volatile matter and the increase in fixed carbon of hydrochar produced at higher HTG temperatures indicated deeper carbonization, probably with higher stability potential when applied to soil.

These findings suggest that if animal manure-based hydrochars can be produced at temperatures near the critical point, but lower than non-catalytic gasification temperatures (i.e., >600 °C), we may produce carbonized hydrochar without losing a significant amount of animal manure C to gas, as Magdengu et al. found with their model compounds. However, there is a lack of studies on HTC of animal manures near critical temperatures and characteristics of the resultant products.

In this study, we hydrothermally carbonized hen and swine manures at 210 °C (a midpoint of the typical subcritical HTC temperature range of 150 to 270 °C) and at 400 °C (slightly higher than the critical point of water) without catalyst addition. We then compared the characteristics of gas, liquid, and solid products from HTC of the two manures at the two temperatures.

## 2. Materials and Methods

### 2.1. Reactor Systems

The subcritical (210 °C) and near-critical (400 °C) HTC of animal manures were carried out in a 500 mL sealed high pressure and temperature reactor made of Alloy C276 with valves and fittings made of T316 Stainless Steel (Model 4575A, Parr Instrument Co., Moline, IL, USA). Animal manure slurry, 20% manure solid (i.e., 100 g water + 25 g dried manure), was heated to 400 °C for 40 min (hereafter referred as near-critical HTC or NCHTC) or 210 °C (HTC) and allowed to cool to room temperature before the reactor was opened. Gas samples were collected from the exhaust valve into a 1 L foil Tedlar bag for analysis. The reactor content was removed and filtered using a 1.6 μm Watman 1820-11 glass fiber filter. The filtered solids (hereafter referred as hydrochar) were dried at 105 °C in an oven. The liquid filtrates were bottled and refrigerated at 4 °C until they were further analyzed.

### 2.2. Feedstock and Product Characteriazation

Swine manure was obtained from a 5600-head finishing swine farm in North Carolina. Laying hen manure was obtained from a large shell egg production farm in Pennsylvania. Hydrochars were dried, and the solid yields (wt %) were calculated from the ratio of grams of dried hydrochar/g dried manure. The hydrochars were characterized for their volatile matter (VM), fixed carbon (FC), ash contents, pH, C, H, N, P, K, and Brunauer–Emmett–Teller (BET) surface area.

The volatile matter, fixed carbon, and ash contents (i.e., proximate analyses) of hydrochar samples were determined using the thermogravimetric method [17]. The BET surface areas of hydrochar samples were measured via $N_2$ adsorption multilayer theory using a Nova 2200e surface area analyzer (Quantachrome, Boynton Beach, FL, USA). The pH value of each hydrochar sample was measured in triplicate at 5% (*w/v*) using deionized water after shaking for 90 min and allowing it to sit for 30 min. The P and K contents of both hydrochar and process liquid samples were determined by X-ray fluorescence (XRF) spectroscopy using a Malvern Panalytical Epsilon 1 (Malvern Panalytical, Westburough, MA, USA). Liquid samples were measured in analytical cups using Mylar film (3.6 μm thickness) as the base. The hydrochars were measured as self-supporting pellets prepared using Hoechest wax C micropowder. Composition was quantified using the Omnian standardless analysis package.

Gross heat of combustion or higher heating value (*HHV*) was measured (triplicates, means reported) using a model C2000 IKA (Wilmington, NC, USA) analytical bomb calorimeter operating in isoperibol mode (30 °C) according to ASTM D4809 [18]. A model G-5012 halogen-resistant decomposition vessel was used and was pressurized (30 mbar) with dry oxygen (99.6%; Gateway Airgas, St. Louis, MO, USA). The instrument was equipped with a D-Neslab RTE 7.0 cooler (23.5 °C). Combustion was attained using paraffin ignition strips without the need for combustion aids. Calibration was performed using benzoic acid as specified in the standard test method, which had a measured gross heat of combustion of 26.37 MJ/kg versus the literature value of 26.46 MJ/kg [19].

CHN analyses of feedstocks and solid products (i.e., hydrochar) were performed on a Leco CHN628 (St Joseph, MI, USA). Prior to analysis, samples were dried under vacuum at 80 °C for 3 h, then ground to smallest particle size possible. Samples (120 mg) were combusted in tin foil cups utilizing a burn profile of 20 s high-flow oxygen followed by 150 s medium-flow and then 30 s high-flow to achieve complete combustion and trapping of products; detection of carbon and hydrogen were by IR, while nitrogen was by thermal conductivity.

The HTC gas samples were analyzed in triplicate on a HP 6890 GC fitted with a thermal conductivity detector (TCD) operating at 220 °C. Gas separation was accomplished using a Poropack Q column (Restec, Bellafonte, PA, USA) with He as the carrier gas. The column was heated as follows: 40 °C for 2 min, ramped to 180 °C (40 °C/min), held for 5 min, ramped to 200 °C (40 °C/min), held for 10 min, and returned to 40 °C. Gas products were quantified versus calibration of a standard gas mixture consisting of ≈3% (*w/v*) each of $H_2$, $CH_4$, $CO$, $CO_2$, $C_2H_6$, and $C_2H_4$ in He (Linde North America Inc., Murray Hill, NJ, USA) and 5% (*w/v*) each of $C_3H_8$ and $C_3H_6$ in He (ILMO Specialty Gases, Jacksonville, IL, USA). Due to the TCD signal damping effect of He on $H_2$, a standard curve for $H_2$ was prepared from mixtures ranging from 3% to 50% (vol/vol), resulting in a calibration curve of $y = 24x^{1.5}$ ($R^2 = 0.997$). For $CO_2$, $C_2H_4$, $C_2H_6$, $C_3H_6$, and $C_3H_8$ in the NCHTC, gas samples were analyzed using GC–MS (Agilent 7890). Gas samples were routed through a GS-CarbonPlot column (30 m long and 0.53 mm id, J&W Scientific, Folsom, CA, USA). Initial oven temperature was 35 °C. After 5 min, the temperature was increased at a rate of 25 °C/min until a final temperature of 250 °C was achieved.

For $H_2$ and $CH_4$ in the NCHTC, gas samples were injected into a GC (HP5890) equipped with a TCD and a Carboxen 1010 Plot column (30 m × 0.53 mm i.d., Supelco, Bellefonte, PA, USA) for determination of $H_2$ and $CH_4$ concentrations (carrier gas was argon). Initial oven temperature was held constant at 35 °C for 7.5 min and subsequently

increased to 240 °C at a rate of 24 °C /min. To quantify CO, a gas sample was injected into a GC (HP5890) equipped with a TCD and a Carboxen 1010 Plot column (30 m × 0.53 mm i.d., Supelco) (carrier gas was helium). Initial oven temperature was held constant at 35 °C for 7.0 min and subsequently increased to 225 °C at a rate of 24 °C/min.

Liquid samples were analyzed using a Shimadzu QP2010 SE GC/MS/FID. The liquid samples were treated with a fourfold excess of $CH_2Cl_2$ to extract the analytes. This solution was separated on a Supelco Petrocol DH 50.2 (50 m × 0.2 mm × 0.5 μm) column in a Shimadzu QP2010 SE GC/MS. The oven program was as follows: initial temperature was held at 50 °C for 3 min, ramped at 20 °C/min to 275 °C, with a final hold time of 2 min. The MS was operated in electron impact (EI) mode at 70 eV.

The chemical oxygen demand (*COD*) of the aqueous filtrates was measured with the closed reflux, colorimetric method (Standard Method 5220 D) according to Standard Methods for the Examination of Water and Wastewater [20]. Total Kjeldahl nitrogen (TKN) was quantified using acid digestion [21] and subsequent analysis using the salicylate method adapted to the microplate format [22].

Prior to anion and cation analysis, the aqueous filtrates were pre-filtered through 0.2 μm nylon syringe filters (Environmental Express, Charleston, SC, USA). After filtration, cation ($NH_4$–N and K) and anion $PO_4$–P concentrations in solution were quantified by chemically suppressed ion chromatography (IC) using ASTM standards D4327-11 and D6919-09 [23] for cations and ASTM D4327-11 [24] for anions. Prior to quantification, all samples were filtered through 0.45μm polyvinylidene fluoride (pvdf) syringe filters. Because of interferences with phosphorus on the IC from organic components of the samples, soluble reactive phosphorus (SRP) was quantified by the malachite green method [25] and further verified using an Agilent 5110 ICP-OES (Agilent Technologies, Santa Clara, CA, USA).

*2.3. Data Interpretation*

The carbon gasification efficiency (*CGE*) was estimated on the basis of the mass of carbon in each of the product gases CO, $CO_2$, $CH_4$, $C_2H_4$, $C_2H_6$, $C_3H_6$, and $C_3H_8$.

$$CGE = \sum_i \frac{mc_i}{mc_o} \tag{1}$$

where $mc_o$ = total *C* in raw feedstock (g);
　　$mc_i$ = total *C* in each product gas *i* (g).
　　The *COD* removal efficiency (*CRE*) was calculated as

$$COD_{RE} = \frac{COD_o - COD_{liq}}{COD_o} \tag{2}$$

where $COD_o$ = initial *COD* in the manure (g $O_2$);
　　$COD_{liq}$ = process liquid *COD* (g $O_2$).
　　The energy recovery (*ER*) was based on the energy recovered in the form of both hydrochar and product gases.

$$ER = \frac{y_g HHV_g + y_{hc} HHV_{hc}}{HHV_o} \tag{3}$$

where $HHV_g$ = higher heating value of product gas (kJ/m$^3$);
　　$HHV_o$ = higher heating value of raw manure (kJ/kg);
　　$HHV_{hc}$ = higher heating value of hydrochar (kJ/kg);
　　$y_{hc}$ = hydrochar yield per kg of raw manure (kg/kg);
　　$y_g$ = product gas yield per kg of raw manure (m$^3$/kg).

### 2.4. Statistical Analysis

The central tendency and precision of measurements were presented with arithmetic averages and standard deviations (given as ± values). All statistical parameters and tests were obtained/performed using GraphPad Prism 9.12 (GraphPad Software, Inc., La Jolla, CA, USA). For comparing multiple sample means, the Brown–Forsythe and Welch ANOVA tests built in the Graphad Prism were performed.

## 3. Results and Discussion

### 3.1. Hydrochar Characteristics

Various physicochemical characteristics of the raw manures and their hydrochars are shown in Table 1. As expected, the hydrochar yield ($y_{hc}$) decreased from 46.2 to 35.5% and from 59.1 to 42.7% for HTC and NCHTC treatments of hen and swine manures, respectively. The VM in both SM and HM hydrochar decreased with HTC temperature, while FC and ash of both HM and SM hydrochar increased with HTC temperature. The FC in HTC-SM and NCHTC-SM hydrochars were higher than that of raw manure and increased with HTC temperature. However, FC of HM hydrochars (both HTC and NCHTC) were not significantly different from that of raw HM ($p = 0.05$). This was unexpected as the FC in both SM and poultry litter hydrochars were significantly higher than that of raw manures [3]. We suspected that higher precipitation of the HM inorganics during the carbonization process caused the high increase in ash and dwarfed the relatively small changes in FC and lower *HHV*.

**Table 1.** Physico-chemical characteristics of raw manures and hydrochars.

| Parameters | Raw HM | HTC-HM | NCHTC-HM | Raw SM | HTC-SM | NCHTC-SM |
|---|---|---|---|---|---|---|
| | | | Proximate Analysis | | | |
| Volatile matter (VM, %$_{db}$) | 65.5 ± 0.1 | 54.7 ± 0.8 | 38.5 ± 1.2 | 68.6 ± 0.2 | 63.7 ± 1.2 | 43.0 ± 1.0 |
| Fixed carbon (FC, %$_{db}$) | 9.3 ± 0.5 | 8.1 ± 0.8 | 9.0 ± 1.3 | 11.9 ± 0.1 | 15.2 ± 1.2 | 22.6 ± 0.3 |
| Ash (%$_{db}$) | 25.3 ± 0.5 | 37.2 ± 0.6 | 52.4 ± 1.4 | 19.6 ± 0.2 | 21.1 ± 0.3 | 34.3 ± 0.8 |
| Hydrochar yield, ($y_{hc}$, %$_{db}$) | - | 46.2 ± 0.6 | 35.5 ± 1.0 | - | 59.1 ± 0.8 | 42.7 ± 1.8 |
| *HHV* (MJ/kg) | 11.6 ± 0.4 | 9.8 ± 1.3 | 9.9 | 18.6 ± 0.2 | 21.3 ± 0.9 | 22.5 |
| Predicted *HHV* (MJ/kg) | 16.2 ± 0.1 | 12.4 ± 0.2 | 11.2 ± 1.1 | 22.9 ± 0.5 | 23.7 ± 0.3 | 23.5 ± 1.5 |
| | | | Ultimate Analysis | | | |
| C (%$_{daf}$) | 47.0 ± 0.6 | 52.8 ± 0.8 | 65.2 ± 3.7 | 57.0 ± 1.1 | 66.0 ± 0.8 | 80.3 ± 4.3 |
| H (%$_{daf}$) | 7.6 ± 0.3 | 5.6 ± 0.1 | 5.0 ± 0.1 | 9.7 ± 0.1 | 8.4 ± 0.0 | 8.3 ± 0.1 |
| N (%$_{daf}$) | 9.7 ± 0.7 | 4.7 ± 0.1 | 4.1 ± 0.2 | 4.1 ± 0.1 | 3.5 ± 0.1 | 3.6 ± 0.1 |
| O (%$_{daf}$) | 35.7 ± 0.8 | 36.9 ± 0.8 | 25.7 ± 3.7 | 29.2 ± 1.2 | 22.2 ± 0.8 | 7.7 ± 4.1 |
| Atomic H/C | 1.93 | 1.26 | 0.92 | 2.05 | 1.53 | 1.24 |
| Atomic O/C | 0.57 | 0.52 | 0.30 | 0.38 | 0.25 | 0.07 |
| | | | Other Analyses | | | |
| P (%$_{db}$) | 0.8 ± 0.1 | 1.5 ± 0.4 | 2.1 ± 1.7 | 1.6 ± 0.3 | 1.6 ± 0.6 | 2.0 ± 0.6 |
| K (%$_{db}$) | 2.3 ± 0.2 | 0.8 ± 0.1 | 0.4 ± 0.1 | 1.7 ± 0.2 | 0.6 ± 0.2 | 0.8 ± 0.2 |
| *COD* (g $O_2$/g solid) | 1.17 ± 0.16 | 0.95 ± 0.09 | 0.77 ± 0.08 | 2.28 ± 0.59 | 1.61 ± 0.24 | 1.28 ± 0.25 |
| pH | 5.7 ± 0.0 | 6.9 ± 0.1 | 7.8 ± 0.4 | 6.5 ± 0.0 | 6.4 ± 0.0 | 6.5 ± 0.2 |

Although FC of HM was minimally impacted by HTC treatment, the elemental compositions of manure and hydrochar changed significantly after HTC treatment. The C content of raw HM (47.0% dry and ash-free, or daf) increased to 52.8% for the HTC-HM and 65.2% for the NCHTC-HM. A similar increase in C content was observed for HTC-SM and NCHTC-SM. In contrast, the oxygen content consistently decreased with HTC treatment. In order to delineate potential HTC reaction pathways, the atomic ratios of H/C and O/C were analyzed using a Van Krevelen diagram (Figure 1). Both O/C and H/C of raw manure decreased with HTC treatment and temperature. The NCHTC-HM and NCHTC-SM yielded lower O/C and H/C than HTC-HM and HTC-SM, respectively. The O/C ratio decreased with HTC temperature as a result of dehydration and decarboxylation. The atomic ratio of H/C decreased with HTC temperature for both HM and SM, potentially due to dehydrogenation and dehydration, suggesting formation of more thermally stable

aromatic groups during carbonization processes. The HTC-HM overlaps the biomass and peat regions, while NCHTC-HM lay over the lignite zone. A similar trend was observed with raw SM, HTC-SM, and NCHTC-SM, although they were slightly above these regions with higher H/C values. This trend indicated that higher HTC temperatures resulted in more carbonized hydrochars.

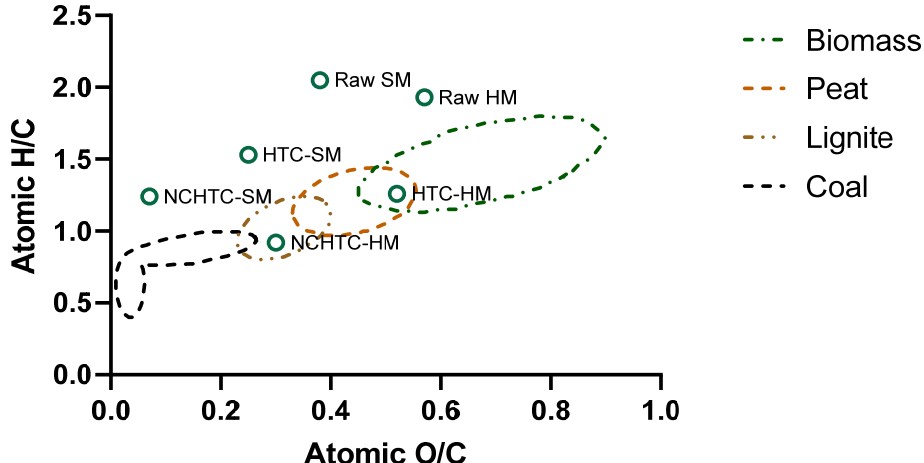

**Figure 1.** Position of raw manures and hydrochars in a Van Krevelen diagram.

The BET surface areas for both raw manures and hydrochars were less than 3.9 m²/g, similar to that of nonactivated biochar [26]. The pH of HM hydrochar slightly increased with HTC treatment from 5.76 to 7.80, while that of SM hydrochar remained the same at about 6.4. The P content for both manures increased with HTC treatment and temperature from 0.8 to 2.1 and 1.1 to 2.0 for HM and SM, respectively. It was reported that P remains in the hydrochar during HTC treatment. In contrast, K content decreased with HTC treatment and temperature for both manures. The decrease in K resulted from dissolution of K from manure to process liquid [27]. Similarly, manure *COD* also decreased with HTC treatment and temperature from 1.17 to 0.77 g $O_2$/g solid and 2.28 to 1.28 g $O_2$/g solid for HM and SM, respectively. The decrease in *COD* probably results from loss of oxygen-demanding organics to $CO_2$ during HTC treatment.

*3.2. Higher Heating Value (HHV) of Hydrochar*

The *HHV*s of HTC-SM and NCHTC-SM hydrochars were higher than that of raw SM as expected, although the *HHV*s of raw SM and HTC-SM were not significantly different ($p = 0.9$). Surprisingly, the *HHV* of HTC-HM decreased with HTC treatment at both temperatures, although the *HHV*s of raw HM and HTC-HM were not significantly different at $p = 0.05$. More statistical comparisons of these hydrochars and raw manures could not be performed due to failure of the bomb calorimeter. Instead, we compared the *HHV*s of these hydrochars and raw manures estimated using the universal *HHV* correlation based on elemental compositions [28]. Because we did not measure S, we treated S contents in raw manures and hydrochars as negligible when using Equation (4). From our previous studies, we found that S in raw manures and hydrochars were generally less than 1% [7]. Therefore, this omission should not significantly impact the calculated *HHV* values.

$$HHV_c = 0.3491\,C + 1.17873\,H + 0.1005\,S - 0.1034\,O - 0.0151\,N - 0.021\,A \qquad (4)$$

where $HHV_c$ = calculated *HHV* of biomass (MJ/kg);
    $C$ = carbon content ($\%_{\_db}$);
    $H$ = hydrogen content ($\%_{\_db}$);
    $S$ = sulfur content ($\%_{\_db}$);
    $O$ = oxygen content ($\%_{\_db}$);

$N$ = nitrogen content ($\%_{\_db}$);

$A$ = ash content ($\%_{\_db}$).

Figure 2 shows the predicted *HHV* against the measured *HHV*s of hydrochar and raw manures. Similar to the measured *HHV*s, the predicted *HHV* of raw HM was still higher than that of both HTC-HM ($p = 0.0015$) and NCHTC-HM ($p = 0.0003$). The predicted *HHV*s of NCHTC-HM and HTC-HM were not significantly different at $p = 0.05$. For SM, there was no significant difference among raw SM, NCHTC-SM, and HTC-SM.

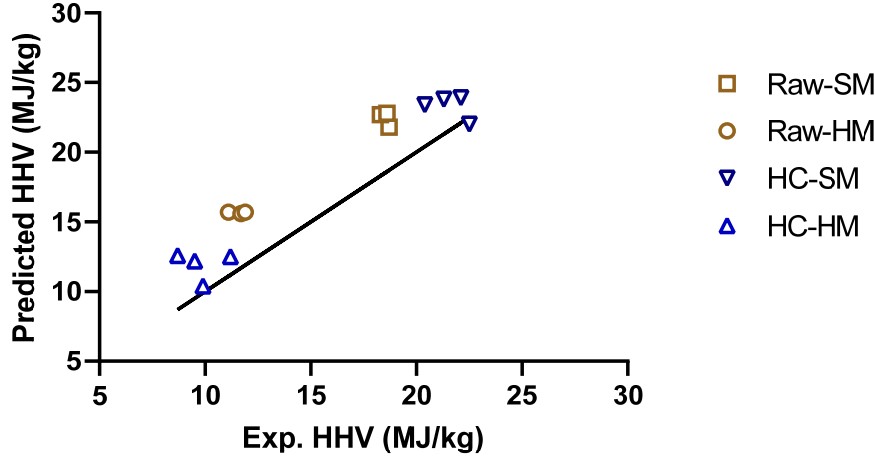

**Figure 2.** *HHV*s of hydrochar and predicted *HHV*s using the correlation by Channiwala and Parikh, 2002 [28].

*3.3. Process Liquid Characteristics*

The pH of HM process liquids ranged from 7.3 to 7.9, which represented an increase from the pH (5.8) of raw HM manure (Table 2). The pH of SM process liquid decreased from 6.3 (raw SM manure) to 5.3, but the pH increased to 7.5 for NCHTC-SM. The increase in pH from HTC to NCHTC process liquid for both manures might have been due to decreases in acetic acid (Table 3) and increases in $NH_4$-N with HTC temperatures. The $NH_4$-N in the process liquid increased from 4.4 to 9.1 g/L and 1.0 to 3.5 g/L for HM and SM HTC and NCHTC treatment, respectively. The levels of $NH_4$-N and TKN for HTC-HM were higher than that from HTC treatment of cow manure [29], but similar to that from HTC treatment of poultry slaughter house sludge cake at 200 and 220 °C [30]. The significant increase in $NH_4$-N while TKN remained relatively constant suggested that more organic compounds were decomposed at higher HTC temperatures, resulting in the conversion of organic N to inorganic N in the liquid as $NH_4$-N. This also corroborated with the decrease in TOC and *COD* with HTC temperature. The *COD* of raw HM decreased from 292 to 93.3 from HTC treatment and was further reduced to 49.7 g-$O_2$/L by NCHTC treatment. Similar trends were observed for SM manures. The TOC also decreased from 31.2 to 16.8 g/L and 18.0 to 9.7 g/L for HTC and NCHTC treatment of HM and SM, respectively. The decrease in *COD* and TOC suggested that more oxygen-demanding organic compounds were removed with the higher temperature HTC treatment. Consequently, the *COD* removal rate ($COD_{RE}$) increased with HTC temperature (Table 2). It was interesting to note that the TOC/COD ratio was consistent and ranged from 0.31 to 0.34. In fact, the *COD*-TOC correlation (i.e., $COD = 49.2 + 3.00 * TOC$) developed by Dubber and Gray [31] for municipal waste fit well for the process liquids (Figure 3). This correlation may provide a convenient way of estimating TOC on the basis of *COD* or vice versa.

**Table 2.** Chemical characteristics of process liquid.

| | pH | *COD* (g-O$_2$/L) | TOC (g/L) | TKN (g/L) | NH$_4$-N (g/L) | PO$_4$-P (mg/L) | *COD$_{RE}$* (%) |
|---|---|---|---|---|---|---|---|
| Raw HM [†] | 5.8 ± 0.1 | 292 ± 40 | - | 15.2 ± 0.4 | 3.1 ± 0.03 | - | - |
| Raw SM [†] | 6.3 ± 0.2 | 570 ± 149 | - | 6.77 ± 2.0 | 0.4 ± 0.01 | - | - |
| NCHTC-HM | 7.9 ± 0.1 | 49.7 ± 2.0 | 16.8 ± 0.3 | 10.1 ± 0.5 | 9.1 ± 0.6 | 2.4 ± 0.4 | 83 ± 12 |
| HTC-HM | 7.3 ± 0.0 | 93.3 ± 3.2 | 31.2 ± 0.2 | 10.4 ± 0.3 | 4.4 ± 0.1 | BDL (<1) | 66 ± 9 |
| NCHTC-SM | 7.5 ± 0.2 | 31.7 ± 0.7 | 9.7 ± 0.3 | 5.0 ± 0.2 | 3.5 ± 0.1 | 11.0 ± 3.2 | 94 ± 25 |
| HTC-SM | 5.3 ± 0.0 | 58.5 ± 0.2 | 18.0 ± 0.2 | 3.9 ± 0.2 | 1.0 ± 0.0 | 1421 ± 11 | 90 ± 24 |

[†] pH, TKN, and NH$_4$-N concentrations in filtered raw manure solution and the total *COD* of raw manure slurry solution.

**Table 3.** GC–MS area percentages of organic compounds detected in process liquid.

| Compounds | HTC-HM | NCHTC-HM | HTC-SM | NCHTC-SM |
|---|---|---|---|---|
| Acetamide | 4.6 | 5.0 | - | - |
| Acetol | - | - | 3.2 | - |
| Acetic acid | 70.9 (12.6 mg/mL) | 25.8 (5.85 mg/mL) | 65.9 (4.40 mg/mL) | 32.1 (4.10 mg/mL) |
| 2-Butanone | - | 11.9 | - | 14.8 |
| 2,6-Dimethylpyrazine | 1.3 | 16.4 | 1.8 | 2.9 |
| 3,6-Dimethyl-2(1H)-pyridinone | | | | 3.9 |
| 5-Dimethylaminopyrimidine | | | | 2.2 |
| Ethylpyrazine | 4.0 | | | |
| 2-Ethyl-5-methylpyrazine | 1.3 | | | |
| 2-Ethyl-6-methylpyrazine | | 2.1 | | |
| Hexanoic acid | | | | 2.7 |
| 1,3,4,6,7,9a-Hexahydro-2H-quinolizine | | | | 3.0 |
| Methylpyrazine | 6.4 | 22.6 | 2.9 | 6.9 |
| N-Methylacetamide | | 3.9 | | |
| 3-Methylbutanoic acid | | | 2.0 | |
| 1-Methyl-2-pyrrolidinone | | 5.1 | | 5.7 |
| 2-Methyl-3-pyridinol | | | | 6.6 |
| 6-Methyl-3-pyridinol | | | | 2.5 |
| Phenol | | | | 4.8 |
| 2-Piperidinone | 2.2 | | 7.2 | 5.2 |
| 2,6-Pyradineamine | 1.9 | | | |
| Pyrazine | 1.4 | | | |
| Pyrimidine | | 3.4 | | |
| 3-Pyridinol | 5.9 | | 17.0 | 4.1 |
| 2-Pyrrolinidone | | | | 2.6 |
| Trimethylpyrazine | | 4.0 | | |

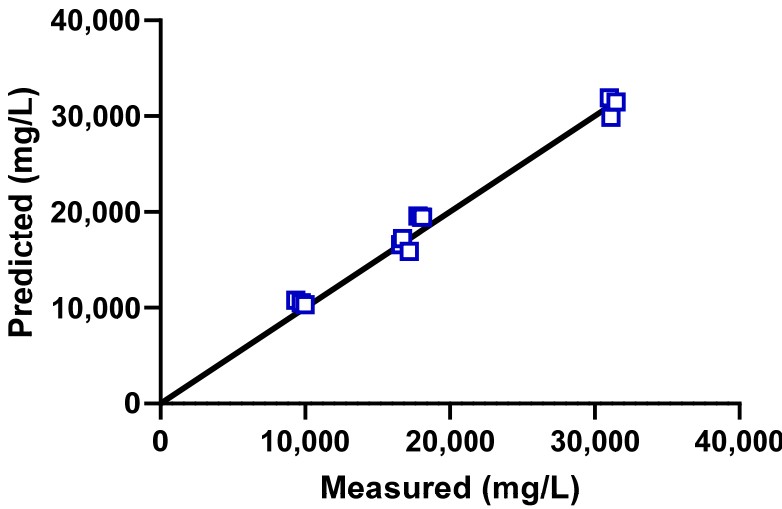

**Figure 3.** Measured DOC compared well with the predicted using correlation by [31].

The $PO_4$-P concentrations in the process liquid were minimal, as other researchers reported that P was mostly incorporated into hydrochar [32]. They claimed the presence of multivalent metal cations such as aluminum, calcium, magnesium, and iron could form insoluble phosphates that were entrapped within the hydrochar. For example, P in the biomass feedstock contributed to ash contents as it precipitated as hydroxyapatite (Ca5(PO4)3OH) [15]. The percent P recovery in the hydrochar ranged from 81.8 to 89.5% and 56.3 to 52.3% for HTC and NCHTC treatment of HM and SM, respectively. The lower P recovery in SM hydrochar was attributed to the lower ash contents of SM hydrochars (Table 1). The large presence of insoluble multivalent metal phosphates in the hydrochar probably contributed to the higher ash contents of the HM hydrochar compared to that of SM (Table 1).

Table 3 shows a number of compounds detected by GC–MS of the process liquid samples. Regardless of HTC temperature, acetic acid was the most dominant organic compound in all process liquids. At higher HTC temperature, the processing liquid contained more diverse organic compounds other than acetic acid, especially for NCHTC-SM. It is interesting to note that the GC–MS area percentages for N-containing organic compounds (i.e., pyrazines, pyridinols, etc.) increased from around one-third for HTC to more than one-half for NCHTC treatment of both HM and SM. Both inorganic N and organic N concentrations in the process liquid increased with HTC temperature. The increase in pH and $NH_4$-N at higher HTC temperature provided the opportunity to extract $NH_4$-N for use as fertilizer. The $NH_4$-N extraction can also assist in the production of biogas via anaerobic digestion of process liquid with low $NH_4$-N concentration [27]. Free ammonia is toxic to anaerobic microorganisms, especially methanogens. Free ammonia concentration as low as 45 mg/L of $NH_3$-N was reported to inhibit the anaerobic digestion process [33]. For example, the free ammonia concentration of the NCHTC-HM process liquid containing 9.1 g/L of NH4-N and pH 7 is about 215 mg/L [34].

### 3.4. Produced Gas Charactersitics

More C in the feedstock was gasified at higher HTC temperatures. The *CGE*s for HTC treatment of HM and SM were 8.9% and 4.2%, respectively (Table 4). The *CGE*s increased to 18 and 12% at higher temperature NCHTC treatment for HM and SM, respectively. The predominant gas in the gaseous products was $CO_2$ for both HTC and NCHTC treatment of HM and SM, while NCHTC treatment produced other energy gases, such as $H_2$, $CH_4$, CO, $C_2H_6$, and other $C_2$ and $C_3$ compounds at trace levels. As a result, the *HHV* value of the produced gases from NCHTC treatment of HM and SM were 6315 and 5816 $kJ/m^3$, respectively. In contrast, the HHC values of produced gases from HTC treatment of HM and SM were only 46 and 91 $kJ/m^3$, respectively. Although the *HHV*s of the produced

gases from NCHTC treatment of HM and SM were much higher than that from HTC treatment, these were still lower than that of produced gases from supercritical HTG or catalytic HTG of wet biomass feedstocks. Almost complete gasification of dairy manure was achieved with produced gas having energy contents ranging from 21.9 to 28.6 MJ/m$^3$ when catalytically gasified at 350 °C [15]. Ro et al. (2007) also estimated that gases with *HHV* of 20.8 MJ/m$^3$ would be produced from catalytic HTG of swine manure at 350 °C [35]. These catalytic HTG processes basically converted all C into gases and produced solid products that mostly consisted of ash. Therefore, NCHTC treatment is desirable if the goal of the treatment is to produce more stable hydrochar despite producing gas with lower energy contents.

**Table 4.** Gas composition, *HHV*, and *CGE* for produced gases.

| | H$_2$ (%) | CO (%) | CH$_4$ (%) | CO$_2$ (%) | C$_2$H$_4$ (%) | C$_2$H$_6$ (%) | C$_3$H$_6$ (%) | C$_3$H$_8$ (%) | *HHV* (kJ/m$^3$) | *CGE* (%) |
|---|---|---|---|---|---|---|---|---|---|---|
| NCHTC-HM | 9.3 ± 2.5 | 2.6 ± 0.6 | 8.1 ± 1.5 | 42.2 ± 2.5 | 0.3 ± 0.0 | 1.3 ± 0.1 | 0.5 ± 0.1 | 0.6 ± 0.1 | 6315 ± 1021 | 18 ± 2.1 |
| HTC-HM | 0.1 ± 0.1 | 0.3 ± 0.0 | 0.0 ± 0.0 | 57.1 ± 1.1 | 0.0 ± 0.0 | 0.0 ± 0.0 | 0.0 ± 0.0 | 0.0 ± 0.0 | 46 ± 7 | 8.9 ± 0.2 |
| NCHTC-SM | 6.4 ± 1.7 | 1.7 ± 0.2 | 5.9 ± 0.2 | 41.2 ± 0.5 | 0.4 ± 0.0 | 1.8 ± 0.0 | 0.6 ± 0.0 | 0.9 ± 0.0 | 5816 ± 130 | 12 ± 0.8 |
| HTC-SM | 0.0 ± 0.0 | 0.7 ± 0.0 | 0.0 ± 0.0 | 43.7 ± 0.1 | 0.0 ± 0.0 | 0.0 ± 0.0 | 0.0 ± 0.0 | 0.0 ± 0.0 | 91 ± 8 | 4.2 ± 0.8 |

*3.5. Recovery of C, N, P, and K*

The distribution of C between solid, liquid, and gaseous products is shown in Figure 4. The C recovery in hydrochar decreased with temperature from 44% and 68% for HTC treatment to 32% and 50% for NCHTC treatment for HM and SM, respectively. The C recovery in process liquid also decreased from 39% and 16% for HTC treatment to 20% and 9% for NCHTC treatment for HM and SM, respectively. In contrast, more gaseous C was produced from NCHTC than HTC for both HM and SM. The C balances for HTC treatment were 87% (HM) and 86% (SM) for HTC treatment but decreased to 61% (HM) and 65% (SM) for NCHTC treatment. The lower C balance suggested that more carbon-containing gases might have been produced from NCHTC treatment than the gas species listed in Table 4.

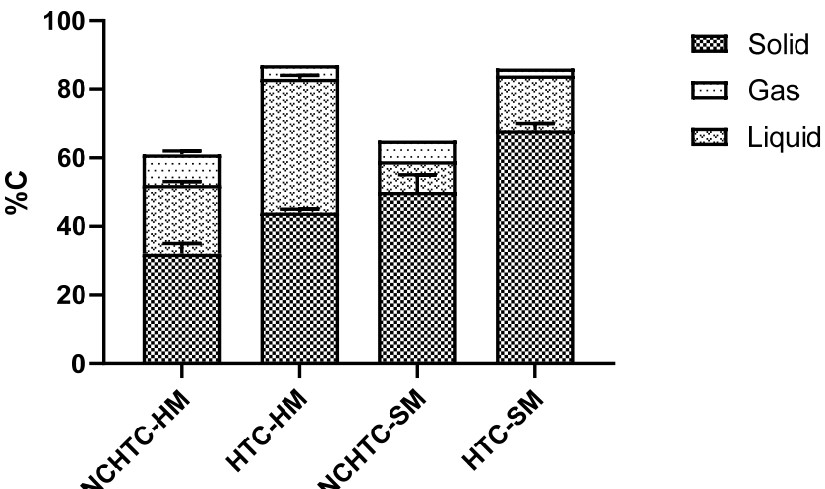

**Figure 4.** Distribution of C in gaseous, liquid, and solid products (Error bars represent standard deviations).

Other than C, which is important for energy and carbon sequestration, other inorganic elements, especially N, P, and K, have high agricultural value, as these can be used as fertilizers. As commercial fertilizer production is energy- and natural-resource-intensive and significantly contributes greenhouse gas emissions to the atmosphere, retaining these elements of the raw manure within the solid and liquid products presents a significant advantage in conserving resources and reducing greenhouse gas emissions. Figure 5 shows the percentage recovery of NPK in solid and liquid products. More N was recovered in

SM hydrochars (49.5% and 31.5% for HTC-SM and NCHTC-SM, respectively) than HM hydrochars (18.7% and 9.5% for HTC-HM and NCHTC-HM, respectively). The N recoveries for process liquid were 47% and 63.6% for HTC-SM and NCHTC-SM, respectively, while 61.1% and 58.3% for HTC-HM and NCHTC-HM, respectively. We saw no indication of volatile amine species in the gas phase. Most K was recovered in the process liquid for HM (89.7% and 80.5% for HTC-HM and NCHTC-HM, respectively); however, 53.2 and 40% for HTC-SM and NCHTC-SM were observed. The K recoveries in hydrochar were lower, ranging from 5.9% to 21.6% due to the high solubility of K in water. As mentioned earlier, most P was recovered in the HM hydrochar, but only about half of P was recovered in SM hydrochar. The P in HTCSM process liquid represented 37.8% of the total P in the raw SM manure. This high P recovery in process liquid might have resulted from the low pH of the HTC-SM liquid. It was not clear why lower pH resulted after HTC treatment of SM.

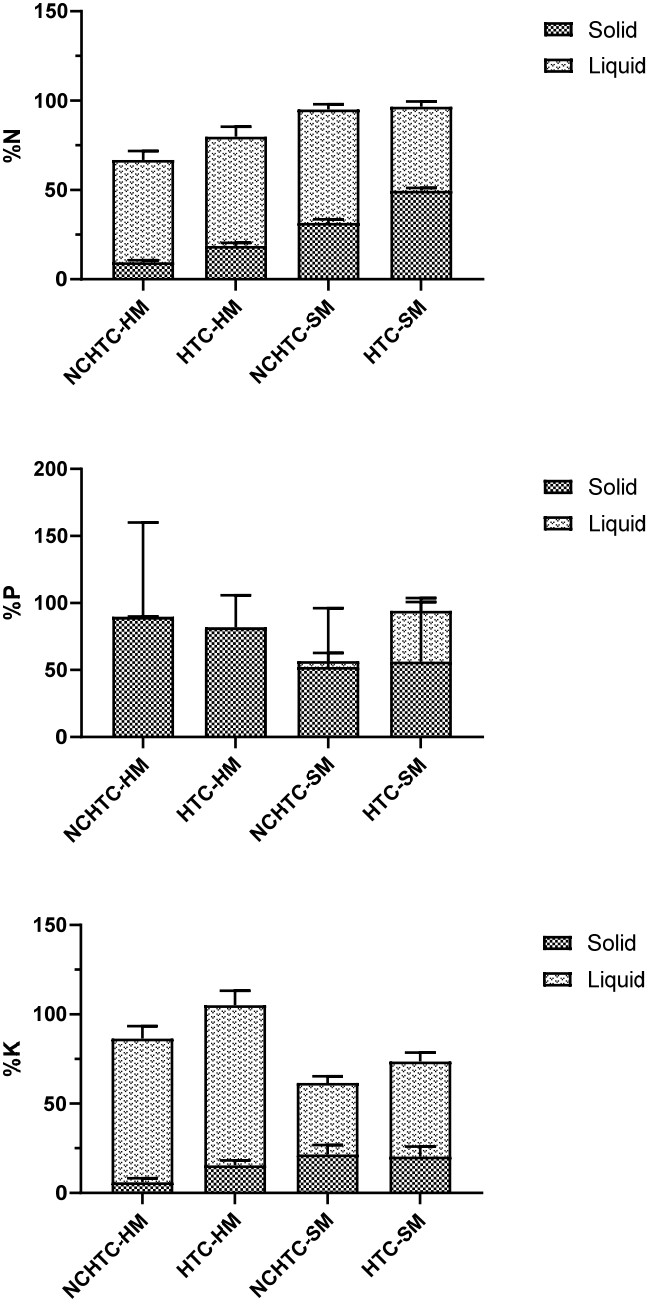

**Figure 5.** Distribution of NPK among solid and liquid products (Error bars represent standard deviations).

*3.6. Heat of Reactions*

The heats of reactions can be estimated by subtracting the sum of the heats of combustion for the products from that of the reactants. The heat of HTC reaction was simplified as follows:

Raw manure hydrochar + process liquid + gas

$$\Delta H_{rxn} = \sum \Delta H_{c,HC} + \Delta H_{c,PL} + \Delta H_{c,g} - \sum \Delta H_{c,manure} \tag{5}$$

where $\Delta H_{rxn}$ = heat of reaction from HTC reaction of 25 g manure (kJ/g, or MJ/kg);

$\Delta H_{c,HC}$ = heat of combustion of hydrochar produced from 25 g manure (kJ);

$\Delta H_{c,PL}$ = heat of combustion of process liquid (kJ);

$\Delta H_{c,g}$ = heat of combustion of produced gas (kJ);

$\Delta H_{c,manure}$ = heat of combustion of 25 g raw manure (kJ).

The heats of combustion for raw manures, hydrochars, and produced gases were calculated from the measured *HHV*s multiplied by 25 g of dry manure. Because we were not able to measure *HHV*s of process liquid using the bomb calorimeter technique, the heat of combustion of the process liquid was roughly estimated by assuming that TOC of the process liquid as the concentration of acetic acid (Table 3). The heats of reaction for both HTC and NCHTC reactions were all exothermic, ranging from −5.7 to −8.6 MJ/kg and more exothermic at higher HTC temperatures (Table 5). These values were slightly larger than or comparable to the values reported in the literature: −1.53 MJ/kg for grape seed [36]; −2.62 MJ/kg for mixed municipal solid waste [37]; −7.3 MJ/kg for the organic fraction of municipal solid waste [38].

**Table 5.** Heat of HTC reaction.

| | Energy in 25 g of Raw Manure (kJ) | Energy in Produced HC (kJ) | Energy in Process Liquid (kJ) [†] | Energy in Produced Gas (kJ) | Energy Recovery (%) | Heat of Reaction (MJ/kg) |
|---|---|---|---|---|---|---|
| NCHTC-HM | 289 | 85.2 | 24.7 | 17.1 | 35 | −7.1 |
| HTC-HM | 289 | 113 | 48.2 | 0.1 | 39 | −6.3 |
| NCHTC-SM | 464 | 229 | 15.8 | 14.3 | 52 | −8.6 |
| HTC-SM | 464 | 314 | 27.8 | 0.1 | 68 | −5.7 |

[†] Estimated from the heat of combustion of acetic acid and its concentration in process liquid.

## 4. Conclusions

Characteristics of solid, liquid, and gaseous products from sub- and near-critical hydrothermal carbonization of HM and SM were evaluated. The higher HTC temperature resulted in more deeply carbonized hydrochar for both HM and SM with significant decrease in VM and increase in ash. For SM, VM and yields decreased, but FC increased with HTC temperature. However, FC of HM did not change significantly with HTC treatment. The *HHV* of HM hydrochar also decreased with HTC treatment, in contrast to the increase in *HHV* for SM with HTC treatment and temperature. Acetic acid was the major organic compound found in the process liquid samples. The TOC and *COD* of process liquids decreased with increasing HTC temperature. This resulted in increased inorganic $NH_4$-N from decomposition of organic-N with increasing HTC temperature. While more than 80% of manure P was recovered in HM hydrochar, slightly more than 50% was recovered in SM hydrochar. Lower ash contents of SM hydrochar might be related to the lower degree of multivalent metal phosphates entrapped in the hydrochar. The NCHTC treatment produced gas with higher energy value but significantly lower *CGE* and *HHV* than that from catalytic HTG of animal manures. Carbon dioxide was the predominant gas for both HTC and NCHTC treatment of HM and SM. The heats of HTC reaction for both HM and SM were exothermic, being slightly more exothermic for NCHTC than HTC treatment. The FC were minimally impacted with HTC treatment, even with NCHTC treatment of HM. However, for SM, the FC approximately doubled while the VM significantly decreased with

a yield of 42.7%, suggesting the high potential for producing more stable hydrochars via NCHTC treatment. Additional work is needed before recommendations on carbonization temperatures can be made. Specifically, there is a need to experimentally investigate how the chars produced from each carbonization condition influence plant growth and soil emissions. In addition, performing a life cycle assessment to evaluate how changes in carbonization conditions influence process environmental impact is desired.

**Author Contributions:** Conceptualization, K.S.R. and M.A.J.; methodology, K.S.R., M.A.J., A.A.S., D.L.C., B.R.M. and N.D.B.; formal analysis, M.A.J., A.A.S., D.L.C., B.R.M. and N.D.B.; data curation, K.S.R. and M.A.J.; writing—original draft preparation, K.S.R.; writing—review and editing, K.S.R., M.A.J., A.A.S., D.L.C., B.R.M. and N.D.B.; supervision, K.S.R.; project administration, K.S.R. All authors have read and agreed to the published version of the manuscript.

**Funding:** This research received no external funding.

**Data Availability Statement:** The data presented in this study will be published and openly available to public in the data repository Dryad (dryad.org) or an equivalent data repository within 30 months after the date of publication and catalogues in the National Agricultural Library (NAL) Ag Data Common with additional information on the USDA funding, data set description, and associated publications.

**Acknowledgments:** We gratefully acknowledge the technical support by Melvin H. Johnson, Jerry H. Martin, II, Tilman Myers, and Paul Shumaker of the Soil, Water and Plant Research Center, and Judy Blackburn, Benetria Banks, and Gary Grose of National Center for Agricultural Utilization Research Center. This research was supported by the United States Department of Agriculture (USDA), Agricultural Research Service (ARS), National Programs 212 Soil and Air and 306 Quality and Utilization of Agricultural Products. Mention of trade names or commercial products in this publication is solely for the purpose of providing specific information and does not imply recommendation or endorsement by the USDA. The USDA is an equal opportunity provider and employer.

**Conflicts of Interest:** The authors declare no conflict of interest.

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
