# Peer review of "Sub- and Near-Critical Hydrothermal Carbonization of Animal Manures"

_sustainability, doi:10.3390/su14095052_

Round 1
Reviewer 1 Report
This work deals with the valorization of animal manures by means hydrothermal carbonization to obtain an hydrochar with improved characteristics to be used in soil amendment/fertility. The manuscript presents an in-depth study on the characteristics of swine and hen hydrochar depending on the carbonization temperatures. However, as the authors also mentioned, additional tests should be done to achieve the claims indicated at the beginning of the manuscript. The following comments address some questions to be considered by the authors.
1.- The abstract should be rewritten in order to answer to the aim of the work. The first lines expose an intention that is not resolved by the results presented.
2.- In the introduction the advantages or disadvantages of hydrochar and biochar from animal manures should be mentioned in a Table, including the aspects related to the process (HTC versus pyrolysis) (Lines 33-40).
3.- In lines 59-73 the authors comment the catalytic and no-catalytic HTG, but after that they do not show interest by catalytic HTG.
4.- Regarding yield of hydrochar, how is possible to carried out the carbonization near the critical point without a significant loss of hydrochar? Examples including yield values should be included (Line 74-78).
5.- Why have the authors selected a 20 % solid waste to carry out the experiments? Why have they considered appropriate to dry the waste before using? Why is the moisture of the raw material? I consider that one of the advantages of the HTC lies using wet wastes, then a justification is required.
6.- Typo mistake ººC (Line 146-150).
7.- The authors should review the equation 3, can yg be expressed in m3/g?
8.- According to FC of HM, can the authors be sure that the carbonization process is carried out? Is this waste appropriate for HTC treatment? The authors represent Van Krevelen diagram, but the low FC content, and consequently, the low HHV can affect to some potential uses of hydrochar.
9.- As the authors know, the BET surface area does not suffer modifications after HTC, but the data shown are not representative of specific area because of the error of the analysis is superior to the data. Then, I recommend the removal of these data from the manuscript. The authors can express their commentaries about this aspect in the text.
10.- The specific unit of COD gO2/gsolid should be appear also in the Table 1.
11.- S has not been measured in the samples. However, the content of this element can affect to HHV and, also, to O. Have the authors planned to carry out the estimations with an estimated value of S content?
12.- The pH of the raw material is measured from the original slurry or from the mixture feed to the reactor? Can they be different?
13.- The authors have made a study of the N content in the process liquid. Can they include information on the N balance between solid, liquid and gas phase?
14.- The authors should clarify the discussion on P and ash content in Lines 285-293. Although ash content of HM is higher than SM, the ash content of the HC is always superior to the raw material, and in all cases very high.
15.- Could the authors provide more potential uses of process liquid? Could the acetic acid extraction be interesting? More information on this aspect is required.
16.- The data of Table 3 are represented in area percentages. Then, if the response of the compounds were different, the obtained conclusions could change. I recommend express the information in “area units”; then the concentrations of acetic acid included in the table could make sense.
17.- The authors conclude that NCHTC treatment “is desirable if the goal of the treatment is to produce more stable hydrochar despite producing gas with lower energy contents”, but can be this statement supported by the global balance (gas, solid and liquid products)? Have the authors analysed N-compounds in the gas phase? Is it possible the existence of Cl compounds in the gas phase? These aspects should be commented. In the case of NCHTC treatment, the C balance indicates errors in the estimation of GCE that can affect to the conclusions of the work. Have the authors estimated the efficiency calculating the C content of the gas by difference (C solid + C liquid)?
18.- Regarding P, K and N content in the HC and liquid phase the authors indicate the interest of their recovery as fertilizer, but have they though about the possibility to concentrate them in the same phase? Speciation is also important. It could be convenient to make easier the use of the HTC products. Moreover, the authors should revise the data from P; error bars are too much big regarding the data.
19.- The authors should review the unit of heat of reaction of equation 5. Why have they selected 25 g as mass unit?
20.- The conclusions are a summary of the experimental results. I recommend pointing out only the most relevant data. The recommendations are interesting but from my point of view they should be removal from this section, because they are not related to the exhibited results.
Reviewer 2 Report
1. It is essential to correct the notation of units. In accordance with applicable standards. 2. Black and white charts will be easier to read. The magnitude of the deviations is not clear. 3. The conducted analyzes are very time-consuming. 4. Methodology well described. Correct reference to standards. 5. The work is carefully written. 6. Very minor remarks have been marked in the text.

Round 2
Reviewer 1 Report
The authors have answered appropriately to all the questions and/or recommendations. The only aspect that I still do not share with them is the one related to the values ​​of the ABET. These are not consistent for the technique. What is the error associated with the reported value? In the response the authors indicate: "The BET surface areas of non-activated hydrochar/biochar are similar in the literature", but it is not an explanation. I recommend including in the manuscript that "all material present BET surface are lower than XX".
Author Response
Dear Reviewer,
As recommended, we deleted the BET surface area data from the Table. The texts now read:
"The BET surface areas for both raw manures and hydrochars were less than 3.9 m2/g, similar to that of nonactivated biochar."
Attached are the revised manuscripts both in MS Word and PDF formats.
Thank you very much.